# Changes in Amino Acid Profiles and Bioactive Compounds of Thai Silk Cocoons as Affected by Water Extraction

**DOI:** 10.3390/molecules26072033

**Published:** 2021-04-02

**Authors:** Chuleeporn Bungthong, Sirithon Siriamornpun

**Affiliations:** Research Unit of Process and Product Development of Functional Foods, Department of Food Technology and Nutrition, Faculty of Technology, Mahasarakham University, Kantarawichai, Mahasarakham 44150, Thailand; chuleeporn.bt@bru.ac.th

**Keywords:** silk protein extraction, sericin, biological properties, water extraction, antioxidants

## Abstract

Silk proteins have many advantageous components including proteins and pigments. The proteins—sericin and fibroin—have been widely studied for medical applications due to their good physiochemical properties and biological activities. Various strains of cocoon display different compositions such as amino-acid profiles and levels of antioxidant activity. Therefore, the objectives of this study were to find a suitable silk protein extraction method to obtain products with chemical and biological properties suitable as functional foods in two strains of *Bombyx mori* silk cocoon (Nangsew strains; yellow cocoon) and *Samia ricini* silk cocoon (Eri strains; white cocoon) extracted by water at 100 °C for 2, 4, 6 and 8 h. The results showed that Nangsew strains extracted for 6 h contained the highest amounts of protein, amino acids, total phenolics (TPC) and total flavonoids (TFC), plus DPPH radical-scavenging activity, ABTS radical scavenging capacity, and ferric reducing antioxidant power (FRAP), anti-glycation, α-amylase and α-glucosidase inhibition. The longer extraction time produced higher concentrations of amino acids, contributing to sweet and umami tastes in both silk strains. It seemed that the bitterness decreased as the extraction time increased, resulting in improvements in the sweetness and umami of silk-protein extracts.

## 1. Introduction

Silk is a natural protein fiber produced from silkworm (*Bombyx mori*) cocoons. It consists of 70–80% of a two-bundled fibrous protein called fibroin and 20–30% of an amorphous matrix of a water-soluble globular protein called ‘sericin’ which functions as a gum to bond the two fibroin filaments together [1]. Fibroin is a water-insoluble protein, consisting of layers of antiparallel β-sheets in the natural fibers [2], two non-polar amino acids, glycine and alanine, are the main amino-acid components of fibroin. The functional properties of fibroin have been utilized in various fields such as textiles, tissue engineering, drug delivery, imaging, and sensing for advanced applications [3]. Sericin is a family of cocoon proteins specifically synthesized in the middle silk gland of the silkworm *Bombyx mori*; sericin constitutes about 20–30% of total cocoon weight. It has been known to possess biological functions, such as antioxidant and anti-tyrosinase activities [4,5,6]. On the other hand, the pigments in silk cocoons also have antioxidant activities. The compounds in these pigments depend on both the food taken by the larvae (mulberry leaves, cassava leaves, etc.) and the silkworm strain involved [7]. Efforts have been made to determine whether the utilization of sericin proteins extracted from cocoon of silkworm may be beneficial for human consumption. One of the physico-chemical properties of sericin is its antioxidant activity. Cocoon shells vary in color, i.e., white, green, yellow and red. The color components coexist and accumulate in the sericin layer of cocoons. Among different colored cocoons, yellow-green cocoon shells contain various flavanol compounds with antioxidant activity [8].

Sericin is a water-soluble protein. When sericin is dissolved in a polar solvent, hydrolyzed in acid or alkaline solutions, or degraded by a protease, the size of the resulting sericin molecules depends on factors such as temperature, pH, and the processing time. Sericin can be extracted from silk by detaching it from the fibroin part. In the past, fibroin was considered to be more useful than sericin in the silk industry. Therefore, prior to silk spinning, the sericin must be removed, using a degumming process; the resulting sericin is discarded in the effluent. Restoration of sericin from the degumming liquor can lighten the load in the effluent, thus reducing the environmental waste load, while also providing a valuable biopolymer with untold profitable properties [9]. During the last decade, sericin has emerged as a valuable commercial resource in many industries, such as those making cosmetics and pharmaceuticals, and food, as well as in the production of many functional biomaterials [10]. Moreover, sericin has also been found to be useful as a degradable biomaterial, a biomedical material and a polymer basis for forming medical articles and functional membranes. Hence, recovery of sericin not only can reduce the environmental impact but also can generate revenue as a natural value-added material in silk production. The Thai traditional extraction method to dissolve sericin from the silk cocoon involves boiling cocoons in hot water at 80–100 °C. However, this process is time consuming. Therefore, an appropriate extraction method is needed to increase the yield and quality of the extracted sericin, while retaining its health-promoting properties with consideration of functional food or nutraceuticals development. In Thailand, there are a number of silk strains cultivated, mostly in North-eastern Thailand. It can be classified into two types, including Thai traditional yellow silk (*Bombyx mori*; indigenous strain) and imported white silk (*Samia ricini*). However, there is lack of information on the chemical composition and health-promoting properties of Thai silk cocoon extracts. 

Therefore, the aim of the study has been to develop an improved silk-protein extraction method for the Nangsew (yellow) and Eri (white) strains, while retaining the chemical and biological properties of the extracted proteins. The investigations included the effects of various extraction times on amino-acid profiles, total phenolic and flavonoids contents and biological properties assessed by different assays. Additionally, the correlations between those bioactive compounds and biological activities were studied for each silk strain. This research is expected to provide information about the important compounds in processed silk protein, thus to develop a natural silk-based additive for functional foods and to provide valuable by-products for consumer use.

## 2. Results and Discussion

### 2.1. The Effect of Extraction Time on the Amino-Acid Contents of Silk-Protein Extracts (SPE)

The SPE of the two strains of silkworm consisted of all ten essential amino acids and eight non-essential amino acids which are shown in Table 1 and Table 2. The content of total essential amino acids in Nangsew strains was higher than in Eri strains (about 3%, compared at six hours). The amino-acid compositions and contents were improved with longer extraction times, except after six hours in both strains. The amino-acid compositions differed between the strains. The three major amino acids of silk cocoons were serine, glycine and threonine, as found in Nangsew strains (mulberry leaf had been eaten) and Eri strains (cassava leaf had been eaten). In mulberry leaf, there were glycine, alanine and serine. The overall composition of acidic amino groups (i.e., aspartic and glutamic acids) in the mulberry strain were greater than that of the basic amino acids [11]. Nevertheless, all authors agree that serine is the most abundant amino acid, followed by aspartic acid and glycine. A further factor relates to the organization of proteins between the cocoon layers: outer, middle and inner. The outer layer is the most soluble, meanwhile the layer adjacent to the fibroin can be only removed with high pressure, high temperature or alkaline compounds [12]. Under these treatments β-sheet structures are degraded and consequently the protein is denatured, thus increasing its water solubility [13]. 

### 2.2. The Effect of Extraction Time on Protein Content 

The longer extraction times provided cocoon extracts with higher protein contents for both strains. The SPE of the Nangsew strains gave the higher protein content when compared with the Eri strains (Table 1 and Table 2). The protein concentration showed an obvious increase with longer extraction times; the longest time of extraction of eight hours provided the highest protein concentrations. Protein solubility is influenced by amino-acid composition and sequence, molecular weight, and the conformation and content of polar and non-polar groups of amino acids. It is important to utilize fully our knowledge of the amino-acid composition and conformation of proteins, including hydrophobic and hydrophilic properties that influence protein solubility. Protein solubility is also affected by environmental factors: ionic strength, type of solvent, pH, temperature, and processing conditions. A relationship between protein solubility and structure has not been demonstrated [14]. Protein-protein interaction in an aqueous medium is accelerated by hydrophobic interactions between the non-polar groups on the protein. When a protein is denatured, secondary and tertiary structures are altered but the peptide bonds of the primary structure between the amino acids are left intact. Since all structural levels of a protein determine its function, the protein can no longer perform its function once it has been denatured. This is in contrast to intrinsically unstructured proteins, which are unfolded in their native state, but are still functionally active and tend to fold upon binding to their biological target [15].

### 2.3. The Effect of Extraction Time on Amino-Acid Contributions to Taste

Each amino acid has its own unique flavor. Alanine, glycine and serine are sweet tasting, while arginine, phenylalanine, histidine, valine, tryptophan, isoleucine and leucine have bitter taste, and glutamic acid and aspartic acid are umami [16]. The findings of the amino-acid content in Table 1 and Table 2 were used to evaluate the distribution of the three taste categories of SPE for both silk strains, indicating the distribution value as shown in Figure 1. The longer extraction time caused the higher concentrations of amino-acid contributions to sweet and umami tastes in both silk strains. It seems that bitterness decreases as the extraction time increases, resulting in improvements in sweetness and umami when comparing Nangsew strains with Eri strains; Eri strains contained greater sweet taste than Nangsew strains. However, a proper sensory evaluation may be needed if this application is used as a drink for humans.

### 2.4. The Effect of Extraction Time on Total Phenolic Content and Total Flavonoid Content

An extraction time of six hours produced the highest total phenolic content (TPC) and total flavonoid content (TFC) when compared to other times, increasing until six hours, after which there was a small but significant decrease (Table 3). SPE of the Nangsew strains and Eri strains for extraction time at six hours provided the TPC values of 79 and 32, and TFC values of 47 and 13 mg GAE/g DW, respectively. The TPC and TFC contents were highest in the Nangsew strains, probably explaining why the Nangsew strains, particularly the yellow-cocoon type, contain significant amounts of pigments, which are primarily associated with carotenoids and flavonoids. These polyphenolic compounds are most commonly known for their antioxidant properties and other diverse biochemical functions, such as anti-tyrosinase, anti-allergy or anti-inflammatory activities [17]. Normally, pigments coexist and accumulate in the layers of the cocoon. The components that give color to silk cocoons are associated with the phenolic compounds from mulberry leaves, the sole food for *B. mori* larvae, whereas the content of cocoon-color components varies depending on the silkworm strains. There are several native strains of silkworm in Thailand with various colors of their cocoon shells. The most common native Thai silk is Nangnoi (yellow cocoon shell), which has been found to contain several flavonoids such as c-prolinyl quercetins, while other Thai silk cocoons have white and yellow–green shells [18]. The types and amounts of flavonoids in silk cocoons have been found to differ genetically [19].

### 2.5. The Effects on DPPH Radical-Scavenging Activity, ABTS^+•^ and FRAP Assay

DPPH radicals are widely used to investigate the radical-scavenging activity of antioxidant compounds. Generally, antioxidant molecules act as H-atom donors to stabilize the DPPH radicals. The scavenging activity of the SPE is indicated by % inhibition. The extraction time after six hours’ extraction provided the highest DPPH radical-scavenging activity when compared to other times; the activity values increased until six hours, after which they decreased (Table 3). The highest % inhibition was found in the Nangsew strains. The Nangsew strain eats mulberry leaves for food; this diet might improve its antioxidant activity properties due to the excellent antioxidant activities of mulberry leaves, including free-radical compounds such as carotenoids, flavonoids, moracins and other compounds present in the leaves [20] and their TPC values [21]. 

The radical scavenging method, using ABTS^•+^, is based on the decrease of the pre-formed radical cation ABTS^•+^ by the addition of an antioxidant. The extent of decolorization of the ABTS^•+^ chromophore, evaluated by using a spectrophotometer at 734 nm, gives the measure of the antioxidant activity of the sample [22]. The findings followed the same trend as for DPPH radical-scavenging activity.

In addition, the FRAP assay was used to investigate the ability of antioxidants toreduce Fe^3+^ to Fe^2+^ [23]. The FRAP values of the silk protein extracts indicated that these two domesticated Thai silks had the highest reducing power equally (Table 3 and Table 4), but that domesticated Thai yellow silk (Nangsew strains) had higher TPC and antioxidant capacity than those of Eri silk. This difference may be due to domesticated silk being composed of specific pigments in their fibers. The main components of silk pigments are carotenoids and flavonoids which correspond to antioxidant activity [24]. Differences of TPC and antioxidant activity in some types may depend on factors such as host plant, environment, weather and habitat.

### 2.6. The Effect of Inhibitory Activities against the Enzymes α-Amylase and α-Glucosidase

SPE have inhibitory activities against the enzymes α-amylase and α-glucosidase. This activity was highest for water extraction at six hours when compared to other times; it increased until 6 h after which it decreased (Table 4). The enzymes α-amylase and α-glucosidase have important roles in catalyzing the hydrolysis of starch into glucose, resulting in the absorption of glucose from the small intestine into the blood stream, reducing the condition of high blood sugar. Many studies report that plants with flavonoid compounds and phenolic compounds from many medicinal plants have effective inhibition of α-amylase and α-glucosidase [25]. Some similar substances, such as myricetin, can be found in cashew leaves. It is thus significant that in addition, SPE (water or enzyme extracted) from Nangsew strains have total flavonoid contents of 49 and 52 mg GAE/g DW respectively, thus capable of causing considerable inhibitory activity against the enzymes α-amylase and α-glucosidase.

### 2.7. The Effect of Anti-AGEs Formation Activity

SPE have the ability of anti-AGEs formation activity after water extraction of six hours; this was the highest when compared to other times and it increased until six hours and then decreased (Table 4). AGEs provide a reaction between sugar and protein causing the protein to behave unusually in some people. For people with diabetes who have up to 50 times higher blood sugar than normal people, the chance of a high glycation reaction causes the protein to malfunction, especially in the case of complications in patients with diabetes [26]. The anti-glycation capacity of numerous medicinal herbs and dietary plants is comparable or even stronger than that of aminoguanidine. Several studies have demonstrated that anti-glycation activity correlates significantly with the phenolic content of the tested plant extracts [27,28]. The SPE of the Nangsew strains with water extraction at six hours has a total flavonoid content of 47 mg RE/g DW and also considerable anti-AGEs formation activity (22% inhibition). 

### 2.8. Correlations

A correlation analysis (Pearson test) was used to make comparisons (as correlation coefficients, r) among and between values for total phenolic content, total flavonoid content, DPPH radical-scavenging activity, ABTS^+•^, FRAP assay, anti-α-amylase, anti-α-glucosidase and anti-glycation of SPE in Nangsew and Eri strains (Table 5 and Table 6). Each parameter obtained from the study demonstrated strong positive correlations between TPC, TFC, DPPH, ABTS^+•^, FRAP, anti-α-amylase, anti-α-glucosidase and anti-glycation. This may be because the bioactive compounds present in the silk extract are capable of scavenging various free radicals and anti-α-amylase, anti-α-glucosidase and anti-glycation. Our findings have also demonstrated that there is a highly significant linear correlation between results for the two strains. These results were in agreement with thise of Benzie and Stezo [29], who found a strong positive correlation between total phenolic content and FRAP assay. Similar results were also found between TPC and DPPH (R^2^ = 0.711) and FRAP (R^2^ = 0.948), respectively in bitter gourd as reported by Kubola [30]. Kumar et al. [31] conducted an evaluation of antioxidant activity and total phenol in different varieties of *Lantana camara* leaves. The extracts showed strong correlations between phenolic content and DPPH, FRAP and ABTS scavenging activities. The highest correlation (r = 0.998, R^2^ = 0.997) was found between total phenolic content and ABTS scavenging assay. In addition, positive correlations between ABTS and TPC (R^2^ = 0.885) and DPPH (R^2^ = 0.999), respectively, in *Gynura procumbens* leaves have been reported by Kaewseejan et al. [32].

The correlations between biological properties and inhibitory activity were found in the research of Quan et al. [33] who investigated the antioxidant and potential diabetic inhibitory properties of *C. tramdenum* bark extracts. Their results also found that antioxidant assays (including DPPH, ABTS, reducing power, and nitric oxide) were strongly correlated with the α-amylase and α-glucosidase inhibitory assay at *p* < 0.05.

## 3. Materials and Methods 

### 3.1. Silk Cocoons 

Nangsew and Eri strains of Thai silk cocoons were collected. The Nangsew strain (yellow cocoons) was obtained from Baan Hua Saban, Phutthaisong District, Buriram Province, Thailand. The Eri strain (white cocoons) came from the Nong Ya Plong Community, Mancha Khiri, Khon Kaen Province, Thailand.

### 3.2. Chemicals and Reagents

All standards were obtained from Sigma–Aldrich Co. (St. Louis, MO, USA), including the essential amino acids: phenylalanine, valine, tryptophan, threonine, isoleucine, methionine, histidine, arginine, lysine and leucine. The non-essential amino acids studied were glycine, glutamic acid, aspartic acid, glutamine, serine, tyrosine, alanine and asparagine. The antioxidant activity determination reagents were obtained from Sigma–Aldrich Co: Folin–Ciocalteu reagent, 2,2-diphenyl-1-picrylhydrazyl (DPPH), 2,4,6-tripiridyl-s-triazine (TPTZ), 2,2′-azino-bis (3-thylbenzthiazoline- 6-sulphonic acid) (ABTS). The solvents and reagents used in the HPLC analysis were purchased from Merck (Darmstadt, Germany). The α-amylase (from *Aspergillus oryzae*), α-Glucosidase (from *Bacillus stearothermophilus*), *p*-nitrophenyl-α-D-glucopyranoside (PNP-G), bovine serum albumin (BSA) and sodium azide were purchased from Sigma–Aldrich Co.

### 3.3. Extraction with Distilled Water 

Extraction was carried out using the method of Sangwong et al. [34]. Silk cocoons were cut into small pieces. Two grams of these pieces were extracted with 200 mL of distilled water at 100 °C for the specified extraction times. The extracts were centrifuged at 5000× *g* for 10 min by centrifuge (Universal 320/320R, Hettich, Boucherville, QC, Canada). The extracts were filtered through Whatman No. 1 paper under vacuum to remove insoluble material, thus producing the silk protein extract (SPE). 

### 3.4. Protein Determination 

Protein determination was determined described by the Bradford assay [35]. SPE (0.5 mL) was added to 1.0 mL of Bradford solution and incubated at room temperature for 5 min. Bovine serum albumin (BSA) was used as a standard reference protein. The absorbance of samples was measured at 595 nm with a visible spectrophotometer (DR 2700™ Portable Spectrophotometer, Hach, Loveland, CO, USA).

### 3.5. Amino-Acid Content by LCMS/MS

Amino-acid analysis involved an LC–MS-MS (Shimadzu LCMS-8030) triple-quadrupole mass spectrometer, in electrospray ionization (ESI) mode, followed by analysis in a Shimadzu HPLC system (Shimadzu, Kyoto, Japan) (Chumroenphat et al. [36]). HPLC analysis involved a flow rate of 0.2 mL/min; the temperature of the column oven was 38 °C and the autosampler was at 4 °C. The mobile phases were prepared (A) aqueous formic acid 0.1% (*v*/*v*) and (B) the methanol-water (50:50); then the mixer was formic acid 0.1% (*v*/*v*). The autosampler needle was purged with methanol before and after aspiration of the sample. The analyses were performed in triplicate.

### 3.6. Total Phenolic Content (TPC)

Total phenolic content in the silk protein extract was analyzed following the Folin-Ciocalteu method [30]. Briefly, 0.3 mL of SPE was mixed with 2.25 mL of 10% Folin-Ciocalteu reagent dissolved in distilled water. After 5 min incubation, 2.25 mL of 6% sodium carbonate (Na_2_CO_3_) solution was added and the mixtures were left to stand for 90 min at room temperature. The absorbance of the solution samples was measured at 725 nm using a spectrophotometer. The TPC in beans was expressed as mg gallic acid equivalents (GAE) per g dry weight (mg GAE/g DW). The analyses were performed in triplicate.

### 3.7. Total Flavonoid Content (TFC)

Total flavonoid contents of SPE were determined according to Kubola and [30], modified. Briefly, 0.5 mL of each sample solution was mixed with 2.25 mL of distilled water and 0.15 mL of 5% NaNO_2_ solution (*w*/*v*). The solution was allowed to stand for 6 min and then 0.3 mL of 10% AlCl_3_ (*w*/*v*) was added to the solution. After 5 min, 0.1 mL of 1 M NaOH (*w*/*v*) solution was added and then the absorbance was measured at 510 nm using a spectrophotometer. Results was expressed as mg rutin equivalents (RE) per g dry weight (mg RE/g DW).

### 3.8. DPPH Radical Scavenging Activity

The antioxidant activities of the SPE were determined as the scavenging activity of the stable 1,1-diphenyl-2-picrylhydrazyl (DPPH) free radical according to Brand-Williams et al. [37]. Silk protein extract 0.1 mL was added to 3 mL of a 0.001 M DPPH in methanol. The mixture was kept for 30 min in the dark at room temperature. Absorbance was determined at 517 nm using a spectrophotometer. The equation for calculating the DPPH inhibition was as follows:Inhibition (%) = [(Abs. control − Abs. sample)/Abs. control] × 100(1)
where the Abs. sample is the absorbance of sample and Abs. control is the absorbance of the control.

### 3.9. Antioxidant Activity by ABTS Assay

The ABTS radical cation method of Re et al. (1999) [22] was modified to evaluate the free radical-scavenging effects. The ATBS working solution was prepared by combining 7.4 mM ABTS and 2.6 mM potassium persulfate at a ratio of 1:1 (*v*/*v*), allowing it to react in the dark for 12 h at room temperature, and then diluting it with deionized water. The absorbance was determined at 734 nm using a spectrophotometer until it reached 1.1 ± 0.02. Then 150 μL of the SPE and 2850 μL of the ATBS working solution were reacted in the dark for 2 h with mixing, and the absorbance at 734 nm using a spectrophotometer; 100% methanol was used as a control. The equation for calculating the inhibition was as Equation (1). 

### 3.10. Ferric Reducing/Antioxidant Power Assay (FRAP)

The reducing power of extracts was based on the FRAP assay [23]. The fresh FRAP reagent was prepared by mixing 100 mL of acetate buffer (0.3 mol L^−1^, pH 3.6), 10 mL of TPTZ solution dissolved in 10 mL of 40 mmol L^−1^ HCl and 10 mL of 20 mmol L^−1^ FeCl_3_ in a ratio of 10:1:1 and 120 mL of distilled water at 37 °C. Briefly, 60 μL of SPE and 180 μL of deionized water were added to 1.8 mL of FRAP reagent. The solution was shaken by vortex mixing. After the solution was incubated in a water bath at 37 °C for 4 min, the absorbance was measured at 593 nm against a control. FRAP values were displayed as mg FeSO_4_ per g dry basis (mg FeSO_4_/g DW).

### 3.11. Inhibitory Activity against the Enzyme α-Amylase

α-Amylase enzyme inhibitory activity was measured using the method of Xiao et al. (2006) [38] with slight modifications. Substrate was prepared by boiling 100 mg potato starch in 5 mL phosphate buffer (pH 7.0) for 5 min, then cooling to room temperature. SPE (50 μL), substrate (300 μL) and 5 μg/mL α-amylase solution (20 μL) were mixed and the solution was incubated at 37 °C for 15 min. The reaction was stopped by adding 50 μL 1 M HCl, and then 50 μL iodine solution was added. The absorbance was measured at 650 nm by a microplate reader. Acarbose was used as a positive control. The inhibition percentage of α-amylase was assessed as Equation (1).

### 3.12. Inhibitory Activity against the Enzyme α-Glucosidase

The α-glucosidase enzyme inhibition assay was performed according to the method of Wang and Zhao [39]. SPE (2 mL dissolved in dimethyl sulfoxide; DMSO) and 0.5 U/mL a-glucosidase (40 mL) were mixed in 120 mL of 0.1 M phosphate buffer (pH 7.0). After 5 min pre-incubation, 5 mM *p*-nitrophenyl-α-D-glucopyranoside solution (40 mL) was added, and the solution was incubated at 37 °C for 30 min. The absorbance was measured at 405 nm by UV/Vis absorbance spectrophotometer microplate reader. The inhibition percentage of α-glucosidase was assessed as Equation (1).

### 3.13. Evaluation of Anti-AGEs Formation Activity 

The inhibitory capacities of AGEs formation of the silk protein and its fractions were measured using the method of Vinson and Howard [40]. The total volume of glycation reaction solution (2.5 mL) was prepared by mixing 500 μL of SPE, 500 μL of 20 mg/mL BSA in phosphate buffer, 500 μL of 0.5 M glucose in phosphate buffer and 1 mL of 0.1 M phosphate buffer at pH 7.4 containing 0.02% (*w*/*v*) sodium azide. This mixture was incubated at 37 °C for 5 days in the dark and then the amount of fluorescent AGEs formed was determined using a fluorescent spectrometer with an excitation wavelength of 330 nm and emission wavelength of 410 nm. The percentage of anti-AGEs formation was calculated based on the resulting fluorescent intensity (FI) using the following equation: % Inhibition = [1 − (FIsample − FIsample blank)/(FIcontrol − FIcontrol blank)] × 100 (2)
where the FIsample is the fluorescent intensity of the sample, FIsample blank is the fluorescent intensity of the sample blank, FIcontrol is the fluorescent intensity of the control and FIcontrol blank is the fluorescent intensity of the sample control blank.

### 3.14. Statistical Analysis

All data were analyzed by the SPSS software (IBM SPSS, Chicago, IL, USA). The results are reported as the mean ± standard deviation (SD) of three replicates and data were analyzed using one-way analysis of variance (ANOVA) with the least significant difference (LSD) test to determine the significance relative to the control and Pearson’s correlation test to assess the correlations among the means. In all cases, *p* < 0.05 was considered significant.

## 4. Conclusions

Our study of the effects of extracting silk protein involved extraction with distilled water at 100 °C for six hours; this procedure provided the highest values of protein content, amino acids, total phenolic content (TPC), total flavonoid content (TFC), DPPH radical-scavenging activity, ABTS radical scavenging capacity assay, FRAP assay, anti-α-amylase, anti-α-glucosidase and anti-glycation when compared to other extraction times. In addition, the SPE of the Nangsew strains provided higher contents of all these nutritionally beneficial substances, when compared with extracts of the Eri strains. The results suggest that Nangsew would be the preferred strains for our ongoing plans to develop a natural additive for functional foods, based on sericin extracted during cocoon treatment, which provides opportunities for an alternative in the production of functional drinks.

## Figures and Tables

**Figure 1 molecules-26-02033-f001:**
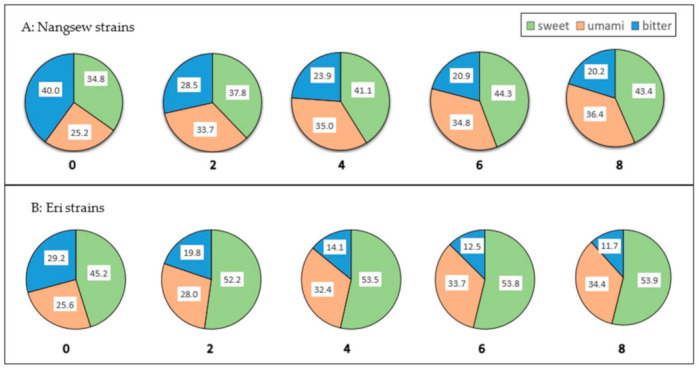
The effect of extraction time (hours) on the contribution of amino acids to the taste of SPE. (Sweet tastes come from alanine, glycine and serine. Umami taste comes from glutamic acid and aspartic acid. Bitter tastes come from arginine, phenylalanine, histidine, valine, tryptophan, isoleucine and leucine.).

**Table 1 molecules-26-02033-t001:** The effect of extraction time on the amino-acid content and protein content of SPE in Nangsew strains.

Parameters			Extraction Time (h)				
			0	2	4	6	8
**Amino acid content** **(µg/g DW)**	Essential amino acids	Phenylalanine	0.63 ± 0.36 ^e^	1.01 ± 0.57 ^d^	1.63 ± 0.64 ^c^	2.44 ± 0.45 ^a^	2.16 ± 0.23 ^b^
		Valine	0.86 ± 0.48 ^e^	2.19 ± 0.11 ^d^	2.69 ± 0.51 ^c^	3.22 ± 0.92 ^a^	2.95 ± 0.55 ^bc^
		Tryptophan	1.07 ± 0.57 ^e^	1.81 ± 0.29 ^d^	2.01 ± 0.09 ^c^	2.79 ± 0.19 ^a^	2.37 ± 0.13 ^b^
		Threonine	1.86 ± 0.62 ^e^	10.38 ± 0.19 ^d^	26.92 ± 0.19 ^c^	29.30 ± 0.46 ^a^	28.61 ± 0.23 ^b^
		Isoleucine	0.96 ± 0.46 ^e^	1.95 ± 0.84 ^d^	2.89 ± 0.94 ^c^	3.91 ± 1.21 ^a^	3.24 ± 0.56 ^b^
		Methionine	0.78 ± 0.54 ^e^	1.48 ± 0.26 ^d^	1.98 ± 0.26 ^c^	2.79 ± 0.65 ^a^	2.21 ± 0.54 ^b^
		Histidine	0.04 ± 0.61 ^d^	7.35 ± 0.17 ^c^	8.92 ± 0.19 ^b^	10.29 ± 0.14 ^a^	8.93 ± 0.26 ^b^
		Arginine	0.97 ± 0.52 ^d^	1.57 ± 0.30 ^c^	1.71 ± 0.33 ^b^	2.64 ± 0.24 ^a^	1.69 ± 0.13 ^b^
		Lysine	1.53 ± 0.41 ^d^	2.44 ± 0.41 ^c^	4.64 ± 0.16 ^b^	5.32 ± 0.31 ^a^	4.88 ± 0.52 ^b^
		Leucine	1.02 ± 0.26 ^d^	1.98 ± 0.65 ^c^	2.31 ± 0.65 ^b^	2.59 ± 0.95 ^a^	2.20 ± 0.15 ^b^
	Total essential amino acids		9.72 ± 0.49 ^e^	32.16 ± 0.66 ^d^	56.50 ± 0.53 ^c^	65.29 ± 0.86 ^a^	59.84 ± 0.48 ^b^
	Non-essential amino acids	Glycine	2.76 ± 0.38 ^e^	11.73 ± 0.38 ^d^	21.32 ± 0.72 ^c^	34.11 ± 0.48 ^a^	30.23 ± 0.52 ^b^
		Glutamic acid	1.93 ± 0.52 ^e^	8.67 ± 0.85 ^d^	10.37 ± 0.53 ^c^	20.67 ± 0.51 ^a^	18.18 ± 0.87 ^b^
		Aspartic acid	1.57 ± 0.48 ^e^	16.44 ± 0.51 ^d^	22.17 ± 0.93 ^c^	25.64 ± 0.23 ^a^	24.25 ± 0.10 ^b^
		Glutamine	0.74 ± 0.63 ^d^	1.87 ± 0.19 ^c^	3.07 ± 0.19 ^b^	3.79 ± 0.46 ^a^	3.01 ± 0.23 ^b^
		Serine	2.01 ± 0.74 ^d^	10.04 ± 0.71 ^c^	15.19 ± 0.69 ^c^	21.60 ± 0.65 ^a^	18.03 ± 1.75 ^b^
		Tyrosine	0.67 ± 0.40 ^e^	1.07 ± 0.37 ^d^	3.27 ± 0.57 ^c^	6.48 ± 0.28 ^a^	5.35 ± 0.06 ^b^
		Alanine	1.06 ± 0.39 ^d^	1.94 ± 0.57 ^c^	2.48 ± 0.27 ^b^	3.25 ± 0.49 ^a^	2.30 ± 0.18 ^b^
		Asparagine	1.96 ± 0.44 ^f^	2.81 ± 1.68 ^e^	3.22 ± 0.98 ^c^	5.06 ± 1.48 ^a^	4.29 ± 0.37 ^b^
	Total amino acids		23.79 ± 0.61 ^e^	91.30 ± 0.64 ^d^	143.38 ± 0.68 ^c^	193.53 ± 0.53 ^a^	172.97 ± 0.60 ^b^
**Protein content (mg/g)**			0.80 ± 0.56 ^e^	1.15 ± 0.42 ^d^	1.24 ± 0.55 ^c^	1.46 ± 0.72 ^b^	2.18 ± 0.23 ^a^

Values are expressed as mean ± standard deviation (*n* = 3). Means with different letters in the row were significant differences at *p* < 0.05.

**Table 2 molecules-26-02033-t002:** The effect of extraction time on the amino-acid content and protein content of SPE in Eri strains.

Parameters			Extraction Time (h)				
			0	2	4	6	8	
**Amino acid content** **(µg/g DW)**	Essential amino acids	Phenylalanine	0.12 ± 0.25 ^e^	0.36 ± 0.67 ^d^	0.52 ± 0.44 ^c^	1.49 ± 0.50 ^a^	0.75 ± 0.32 ^b^	
		Valine	0.73 ± 0.36 ^e^	2.02 ± 0.11 ^d^	2.43 ± 0.51 ^c^		2.61 ± 0.55 ^bc^	
		Tryptophan	0.07 ± 0.31 ^e^	0.41 ± 0.37 ^d^	0.50 ± 0.35 ^c^	0.95 ± 0.20 ^a^	0.63 ± 0.47 ^b^	
		Threonine	1.31 ± 0.69 ^d^	2.08 ± 0.39 ^c^	3.52 ± 0.19 ^b^	4.53 ± 0.51 ^a^	3.37 ± 0.49 ^b^	
		Isoleucine	0.11 ± 0.57 ^e^	0.46 ± 0.77 ^d^	0.53 ± 0.38 ^c^	0.87 ± 0.76 ^a^	0.62 ± 0.42 ^b^	
		Methionine	0.23 ± 0.60 ^e^	0.57 ± 0.30 ^d^	0.68 ± 0.37 ^c^	0.86 ± 0.42 ^a^	0.72 ± 0.49 ^b^	
		Histidine	0.11 ± 0.32 ^e^	0.21 ± 0.36 ^d^	0.54 ± 0.79 ^c^	0.76 ± 0.36 ^a^	0.57 ± 0.67 ^b^	
		Arginine	0.18 ± 0.45 ^d^	0.28 ± 0.62 ^c^	0.39 ± 0.37 ^b^	0.54 ± 0.46 ^a^	0.43 ± 0.25 ^b^	
		Lysine	1.64 ± 0.62 ^e^	2.31 ± 0.35 ^d^	3.08 ± 0.32 ^c^	4.71 ± 0.64 ^a^	2.58 ± 0.48 ^b^	
		Leucine	1.02 ± 0.61 ^e^	1.56 ± 0.44 ^d^	2.07 ± 0.92 ^c^	3.65 ± 0.38 ^a^	2.54 ± 0.47 ^b^	
	Total essential amino acids		5.52 ± 0.51 ^e^	10.26 ± 0.63 ^d^	14.26 ± 0.86 ^c^	21.23 ± 0.72 ^a^	14.82 ± 0.53 ^b^	
	Non-essential amino acids	Glycine	1.44 ± 0.65 ^e^	11.03 ± 0.38 ^d^	19.36 ± 0.43 ^c^	28.66 ± 0.51 ^a^	27.32 ± 0.38 ^b^	
		Glutamic acid	2.51 ± 0.41 ^e^	8.53 ± 0.69 ^d^	12.09 ± 0.53 ^c^	14.03 ± 0.42 ^a^	13.98 ± 0.37 ^b^	
		Aspartic acid	0.54 ± 0.32 ^e^	10.21 ± 0.86 ^d^	10.42 ± 0.73 ^c^	12.87 ± 0.47 ^a^	12.23 ± 0.32 ^b^	
		Glutamine	0.79 ± 0.37 ^e^	1.08 ± 0.56 ^d^	1.32 ± 0.29 ^c^	1.57 ± 0.39 ^a^	1.09 ± 0.75 ^b^	
		Serine	1.24 ± 0.55 ^e^	11.76 ± 0.32 ^d^	13.87 ± 0.43 ^c^	17.62 ± 0.73 ^a^	15.42 ± 1.49 ^b^	
		Tyrosine	1.97 ± 0.53 ^e^	2.37 ± 0.43 ^d^	4.08 ± 0.72 ^c^	5.64 ± 0.43 ^a^	2.79 ± 0.38 ^b^	
		Alanine	0.94 ± 0.47 ^e^	1.09 ± 0.39 ^c^^d^	1.27 ± 0.44 ^c^	1.46 ± 0.37 ^a^	1.17 ± 0.45 ^b^	
		Asparagine	1.07 ± 0.56 ^f^	1.50 ± 0.79 ^e^	1.84 ± 0.37 ^d^	2.10 ± 0.96 ^b^	1.97 ± 0.54 ^c^	
	Total amino acids		16.10 ± 0.58 ^e^	57.98 ± 0.69 ^d^	78.73 ± 0.62 ^c^	105.96 ± 0.68 ^a^	90.86 ± 0.62 ^b^	
**Protein content (mg/g)**			0.81 ± 0.36 ^e^	1.02 ± 0.22 ^d^	1.16 ± 0.31 ^c^	1.21 ± 0.30 ^b^	1.87 ± 0.43 ^a^	

Values are expressed as mean ± standard deviation (*n* = 3). Means with different letters in the row were significant differences at *p* < 0.05.

**Table 3 molecules-26-02033-t003:** The effect of extraction time on total phenolic content, total flavonoid content, DPPH radical-scavenging activity, ABTS^+•^ assay and FRAP assay of SPE.

Strain	Extraction Time (h)	Total Phenolic Content (mg GAE/g DW)	Total Flavonoid Content (mg RE/g DW)	Dpph Radical-Scavenging Activity (% inhibition)	ABTS^+•^ (% Inhibition)	FRAP Assay (mg FeSO_4_/g DW)
Nangsew	0	10.51 ± 1.48 ^e^	5.94 ± 1.63 ^e^	10.51 ± 1.48 ^d^	2.81 ± 0.08 ^e^	1.24 ± 0.13 ^d^
	2	43.03 ± 1.05 ^f^	35.74 ± 1.25 ^d^	43.03 ± 1.05 ^e^	7.86 ± 0.15 ^d^	3.16 ± 0.24 ^c^
	4	58.23 ± 1.05 ^c^	43.03 ± 1.05 ^c^	58.23 ± 1.05 ^c^	15.99 ± 0.26 ^c^	4.59 ± 0.32 ^b^
	6	78.89 ± 1.82 ^a^	47.29 ± 1.11 ^a^	78.89 ± 1.82 ^a^	23.94 ± 0.24 ^a^	5.01 ± 0.22 ^a^
	8	76.28 ± 1.05 ^b^	45.06 ± 1.10 ^b^	76.28 ± 1.05 ^b^	22.87 ± 0.31 ^b^	4.53 ± 0.13 ^b^
Eri	0	10.22 ± 1.11 ^e^	1.57 ± 1.01 ^e^	1.03 ± 0.14 ^e^	1.58 ± 0.11 ^d^	1.02 ± 0.17 ^e^
	2	18.12 ± 1.20 ^d^	8.98 ± 1.17 ^d^	2.28 ± 0.18 ^d^	3.25 ± 0.20 ^c^	2.11 ± 0.24 ^d^
	4	22.54 ± 1.03 ^c^	10.43 ± 1.32 ^c^	5.21 ±0.2c ^c^	4.20 ± 0.26 ^b^	2.68 ± 0.16 ^c^
	6	32.06 ± 1.14 ^a^	12.67 ± 1.13 ^a^	12.36 ± 0.19 ^a^	4.88 ± 0.17 ^a^	3.45 ± 0.22 ^a^
	8	30.08 ± 1.07 ^b^	11.85 ± 1.21 ^b^	11.33 ± 0.13 ^b^	4.19 ± 0.28 ^b^	3.37 ± 0.19 ^b^

Values are expressed as mean ± standard deviation (*n* = 3). Means with different letters in the column within the same strains were significantly different at the level *p* < 0.05.

**Table 4 molecules-26-02033-t004:** The effect of extraction time on inhibitory activity against enzyme α-amylase, α-glucosidase and anti-AGEs formation activity of SPE.

Strain	Extraction Time (h)	Inhibitory Activity against Enzyme α-Amylase (% Inhibition)	Inhibitory Activity against Enzyme α-Glucosidase (% Inhibition)	Anti-AGEs Formation Activity (% Inhibition)
Nangsew	0	2.81 ± 0.08 ^e^	1.24 ±0.13 ^d^	2.17 ± 0.12 ^e^
	2	7.86 ± 0.15 ^d^	3.16 ± 0.04 ^c^	4.68 ± 0.11 ^d^
	4	15.99 ± 0.26 ^c^	4.59 ± 0.02 ^b^	14.66 ± 0.09 ^c^
	6	24.94 ± 0.24 ^a^	5.51 ± 0.02 ^a^	21.90 ± 0.07 ^a^
	8	22.87 ± 0.31 ^b^	4.53 ± 0.03 ^b^	18.88 ± 0.04 ^b^
Eri	0	1.06 ± 0.15 ^e^	0.69 ± 0.51 ^e^	1.58 ± 0.83 ^e^
	2	3.60 ± 0.20 ^d^	1.15 ± 0.37 ^d^	3.46 ± 0.97 ^d^
	4	5.65 ± 0.19 ^c^	2.29 ± 0.26 ^c^	5.55 ± 0.48 ^c^
	6	7.91 ± 0.30 ^a^	3.63 ± 0.35 ^a^	9.30 ± 0.73 ^a^
	8	6.03 ± 0.27 ^b^	3.11 ± 0.41 ^b^	8.56 ± 0.69 ^b^

Values are expressed as mean ± standard deviation (*n* = 3). Means with different letters in the column within the same strain were significantly different at the level *p* < 0.05.

**Table 5 molecules-26-02033-t005:** Correlations among TPC, TFC, DPPH, ABTS^+•^, FRAP assay, anti-α-amylase, anti-α-glucosidase and anti-glycation of SPE in Nangsew strains.

	TPC	TFC	DPPH	ABTS^+•^	FRAP	Anti-α-Amylase	Anti-α-Glucosidase	Anti-Glycation
TPC	1	0.950 **	0.903 **	0.972 **	0.961 **	0.951 **	0.843 **	0.927 **
TFC		1	0.839 **	0.865 **	0.948 **	0.848 **	0.777 **	0.828 **
DPPH			1	0.940 **	0.926 **	0.873 **	0.875 **	0.980 **
ABTS^+•^				1	0.927 **	0.965 **	0.868 **	0.975 **
FRAP					1	0.922 **	0.906 **	0.907 **
Anti-α-amylase						1	0.905 **	0.913 **
Anti-α-glucosidase							1	0.865 **
Anti-glycation								1

TPC: Total phenolic content, TFC: Total flavonoid content, DPPH: 1,1-diphenyl-2-picrylhydrazyl radical. Scavenging activities, FRAP: Ferric reducing antioxidant activities. ** Significant differences at *p* < 0.01.

**Table 6 molecules-26-02033-t006:** Correlation among TPC, TFC, DPPH, ABTS^+•^, FRAP assay, anti-α-amylase, anti-α-glucosidase and anti-glycation of SPE in Eri strains.

	TPC	TFC	DPPH	ABTS^+•^	FRAP	Anti-α-Amylase	Anti-α-Glucosidase	Anti-Glycation
TPC	1	0.923 **	0.965 **	0.925 **	0.982 **	0.994 **	0.977 **	0.974 **
TFC		1	0.795 **	0.993 **	0.971 **	0.880 **	0.962 **	0.855 **
DPPH			1	0.804 **	0.909 **	0.986 **	0.909 **	0.980 **
ABTS^+•^				1	0.973 **	0.888 **	0.973 **	0.874 **
FRAP					1	0.963 **	0.990 **	0.946 **
Anti-α-amylase						1	0.961 **	0.986 **
Anti-α-glucosidase							1	0.959 **
Anti-glycation								1

TPC: Total phenolic content, TFC: Total flavonoid content, DPPH: 1,1-diphenyl-2-picrylhydrazyl radical; scavenging activities, FRAP: Ferric reducing antioxidant activities. ** Significant differences at *p* < 0.01.

## Data Availability

Not applicable.

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
