# Peer review of "Changes in Amino Acid Profiles and Bioactive Compounds of Thai Silk Cocoons as Affected by Water Extraction"

_molecules, 2021, doi:10.3390/molecules26072033_

Round 1
Reviewer 1 Report
In this manuscript Chuleeporn Bungthong and Sirithon Siriamornpun are talking about Process optimization of protein extraction from Thai silk co-coons with respect to amino-acid profiles and bioactive com-pounds. The subject of the article is topical and the article may be useful to other researchers and scientists. Most of the cited articles is from last 20 years. 33 references were cited. The article is actual and actual references were cited.
After publication, few corrections should be prepared:
- The extracts were centrifuged (10 min, 5,000×g). – unclear – authors should use SI units
- Authors did not provide the necessary information about equipment: centrifuge, spectrophotometer
- Equation no 1 should be prepared as a function in Word - insert equation
- The same for equation 2, 3, 4, 5
- Authors should, clarify what was the limitations of this study?
- And also in conclusions, authors should clarify the future perspectives of these results
- Additionally, I think that introduction should be a little bit improved. It is too short, the authors only cite 5 sources in the introduction section. Authors should study more articles about fibroin and expand the introduction section.
Thank you

Author Response
Thank you the reviewer for the comments and suggestions. We have taken all the comments into account point by point as attached. Please also be noted that the title has been changed as suggested by one of the reviewers to "Changes in amino acid profiles and bioactive compounds of Thai silk cocoons as affected by water extraction."
Response to Reviewer 1 Comments
Point 1: The extracts were centrifuged (10 min, 5,000×g). – unclear – authors should use SI units
Response 1: Thank you for your suggestion. We have added more information as suggested. Please see section 3.3.
Point 2: Authors did not provide the necessary information about equipment: centrifuge, spectrophotometer
Response 2: Thank you for your suggestion. We have added more information on equipment as suggested. Please see sections 3.3 and 3.4.
Point 3: Equation no 1 should be prepared as a function in Word - insert equation The same for equation 2, 3, 4, 5
Response 3: Thank you for your suggestion. We have revised equation 1 as suggested. Please see section 3.8.
Point 4: Authors should, clarify what was the limitations of this study?
Response 4: The limitation of this study is sensory evaluation could be done to confirm the overall consumer acceptance of the extract if applied to functional drinks.
Point 5: Additionally, I think that introduction should be a little bit improved. It is too short, the authors only cite 5 sources in the introduction section. Authors should study more articles about fibroin and expand the introduction section.
Response 5: Thank you for your suggestion. We have revised the introduction as suggested. Please see on Introduction.

Reviewer 2 Report
Present research by Bungthong and Siriamornpun investigates protein extraction from Thai silk cacoons in order to determine influence of extraction time on chemical profile. Conventional extraction procedure was used and extraction time was varied at five levels in order to determine protein content, amino-acid profile, polyphenols content and bioactivity determined by various antioxidant and enzyme-inhibitory assays. The study is clearly presented and overall quality of the text is good, however, I have some issues considering novelty, methodology and discussion which should be improved.
Title
- Process optimization should not be used in title since only one parameter was varied in extraction. Please reformulate.
Introduction
- State of the art in silk protein extraction should be added in details with clear pros and cons.
- This should be followed by clearly stated novelty of your work.
Materials and methods
- Information about the sample is missing. Please provide thorough information about sample origin, properties, processing, etc.
- Why did you use conventional extraction procedure? What do you improve comparing to the reference that you cite?
- How and why did you choose extraction time as input factor? What about other extraction parameters? Please give thorough information about selection of experimental plan with appropriate references.
- Please use original references for antioxidant and enzyme-inhibitory assays.
Results and discussion
- Literature comparison is missing is some cases (for example phenols and flavonoids).
- 8. Section should be improved. Please improve discussion for correlations and support it with the appropriate literature.
Conclusions
- Please provide insight of the potential application of the products you obtained. What are the challenges and opportunities?
Author Response
Thank you the reviewer for the comments and suggestions. We have taken all the comments into account point by point as attached. Please also be noted that the title has been changed as suggested by one of the reviewers to "Changes in amino acid profiles and bioactive compounds of Thai silk cocoons as affected by water extraction."
Response to Reviewer 2 Comments
Point 1: Title
Process optimization should not be used in title since only one parameter was varied in extraction. Please reformulate.
Response 1: Thank you for your suggestion. We have changed the title is Changes in amino acid profiles and bioactive compounds of Thai silk cocoons as affected by water extraction.
Point 2: Introduction
State of the art in silk protein extraction should be added in details with clear pros and cons. This should be followed by clearly stated novelty of your work.
Response 2: Thank you for your suggestion. We have revised the introduction as suggested. Please see on Introduction.
Point 3: Materials and methods
Information about the sample is missing. Please provide thorough information about sample origin, properties, processing, etc.
Response 3: In fact, the details of samples are described as in Section 3.1 and the sample preparation is in Section 3.3.
Point 4: Materials and methods
Why did you use conventional extraction procedure? What do you improve comparing to the reference that you cite?
Point 5: Materials and methods
How and why did you choose extraction time as input factor? What about other extraction parameters? Please give thorough information about selection of experimental plan with appropriate references.
Response 4 and 5: We designed the experiment according to the traditional procedure since we want to study how the chemical properties of silk extract change. We expected to use the findings information for practical use for Thai silk users.
Point 6: Materials and methods
Please use original references for antioxidant and enzyme-inhibitory assays.
Response 6: Thank you for your suggestion. We have revised the references as suggested. Please see sections 3.8, 3.9, 3.10, 3.11, and 3.12.
Point 7: Results and discussion
Literature comparison is missing is some cases (for example phenols and flavonoids).
Response 7: In fact, the details of total phenolic and flavonoid are described as in section 2.4.
Point 8: Section should be improved. Please improve discussion for correlations and support it with the appropriate literature.
Response 8: Thank you for your suggestion. We have revised the discussion as suggested. Please see section 2.8.
Point 9: Conclusions
Please provide insight of the potential application of the products you obtained. What are the challenges and opportunities?
Response 9: Thank you for your suggestion. We have revised the conclusions as suggested. Please see on Conclusions.

Reviewer 3 Report
The study aimed to find a suitable silk protein extraction method from Bombyx mori and Samia ricini silk cocoon to obtain products eligible as functional foods. The research was overall well conducted. I have the following minor observations:
- Abstract: please specify what FRAP means;
- Introduction: the antioxidant effect is not necessarily linked only to the pigments. Bombyx mori (white) sericin has also been shown to have antioxidant activity. See for example https://doi.org/10.1016/j.jpba.2020.113291, https://doi.org/10.3892/br.2014.244
- Introduction: some references should be added when writing about the recovery of sericin from wastewater. See for example https://doi.org/10.1002/jctb.6441
- Introduction: when reporting the study’s aim, it is unclear why Bombyx mori and Samia ricini have been used. Please specify. Moreover, here the experimental design should be explained, reporting which analyzes are carried out and why. In this way, the reader has an idea of what he will read in section 2
- Materials and methods: I did not understand the reason for doing three analyzes that give the same information about the antioxidant properties. This must be explained
- Section 3.14: please specify the software used
Author Response
Thank you the reviewer for the comments and suggestions. We have taken all the comments into account point by point as attached. Please also be noted that the title has been changed as suggested by one of the reviewers to "Changes in amino acid profiles and bioactive compounds of Thai silk cocoons as affected by water extraction."
Response to Reviewer 3 Comments
Point 1: Abstract: please specify what FRAP means
Response 1: Thank you for your suggestion. We have added more information about FRAP. Please see the abstract.
Point 2: Introduction: the antioxidant effect is not necessarily linked only to the pigments. Bombyx mori (white) sericin has also been shown to have antioxidant activity. See for example https://doi.org/10.1016/j.jpba.2020.113291, https://doi.org/10.3892/br.2014.244
Response 2: Thank you for your suggestion. We have revised the introduction as suggested. Please see on Introduction.
Point 3: Introduction: some references should be added when writing about the recovery of sericin from wastewater. See for example https://doi.org/10.1002/jctb.6441
Response 3: Thank you for your suggestion. We have revised the introduction as suggested. Please see on Introduction.
Point 4: Introduction: when reporting the study’s aim, it is unclear why Bombyx mori and Samia ricini have been used. Please specify. Moreover, here the experimental design should be explained, reporting which analyzes are carried out and why. In this way, the reader has an idea of what he will read in section 2
Response 4: Thank you for your suggestion. We have revised the introduction as suggested. Please see on Introduction.
Point 5: Materials and methods: I did not understand the reason for doing three analyzes that give the same information about the antioxidant properties.
Response 5: The reason for using more than one analysis for determining the antioxidant properties is that different methods (or assays) to chemical reaction differently. Since there are numerous antioxidant compounds in biomaterials and they have different antioxidant properties. So, we used three different methods to evaluate them as follows:
1) FRAP assay is a method for measuring total reducing power of electron donating substances.
2) DPPH assay is a method for measuring the ability of antioxidant molecules to quench DPPH free radicals.
3) ABTS assay is a method used to determine the antioxidant capacity of food products.
Point 6: This must be explained Section 3.14: please specify the software used.
Response 6: Thank you for your suggestion. We have added the SPSS software in Section 3.14 as suggested.

Round 2
Reviewer 1 Report
The article can be published in the current form